DATA RELEASE

# AIMSurv: First pan-European harmonized surveillance of *Aedes* invasive mosquito species of relevance for human vector-borne diseases

Miguel Ángel Miranda[1,*], Carlos Barceló[1], Daniele Arnoldi[2], Xenia Augsten[3], Karin Bakran-Lebl[4], George Balatsos[5], Mikel Bengoa[6], Philippe Bindler[7], Kristina Boršová[8], Maria Bourquia[9], Daniel Bravo-Barriga[10], Viktória Čabanová[8], Beniamino Caputo[11], Maria Christou[12], Sarah Delacour[13], Roger Eritja[14], Ouafaa Fassi-Fihri[50], Martina Ferraguti[15], Eleonora Flacio[16], Eva Frontera[10], Hans-Peter Fuehrer[17], Ana L. García-Pérez[18], Pantelis Georgiades[12], Sandra Gewehr[19], Fátima Goiri[18], Mikel Alexander González[20], Martin Gschwind[21,52], Rafael Gutiérrez-López[1], Cintia Horváth[22], Adolfo Ibáñez-Justicia[23], Viola Jani[24], Përparim Kadriaj[24], Katja Kalan[25], Mihaela Kavran[26], Ana Klobucar[27], Kornélia Kurucz[28], Javier Lucientes[13], Renke Lühken[29], Sergio Magallanes[15], Giovanni Marini[2], Angeliki F. Martinou[30], Alice Michelutti[31], Andrei Daniel Mihalca[22], Tomás Montalvo[32], Fabrizio Montarsi[31], Spiros Mourelatos[19], Nesade Muja-Bajraktari[33], Pie Müller[21,52], Gregoris Notarides[34], Hugo Costa Osório[35], José A. Oteo[36], Kerem Oter[37], Igor Pajović[38], John R. B. Palmer[39], Suncica Petrinic[27], Cristian Răileanu[40], Christian Ries[41], Elton Rogozi[24], Ignacio Ruiz-Arrondo[36], Isis Sanpera-Calbet[39], Nebojša Sekulić[42], Kivanc Sevim[43], Kurtesh Sherifi[44], Cornelia Silaghi[40], Manuel Silva[35], Nikolina Sokolovska[45], Zoltán Soltész[46], Tatiana Sulesco[47], Jana Šušnjar[25], Steffanie Teekema[23], Andrea Valsecchi[32], Marlen Ines Vasquez[34], Enkelejda Velo[51], Antonios Michaelakis[5], William Wint[48], Dušan Petrić[26], Francis Schaffner[49], Alessandra della Torre[11] and Consortium AIM-COST/AIM-Surv[†]

**Submitted:** 23 March 2022

\* Corresponding author. E-mail: ma.miranda@uib.es

† Collaborative Authors and their affiliations appears at the end of the publication

Preprint submitted at https://doi.org/10.5281/zenodo.6394647

Included in the series: *Vectors of human disease* (https://doi.org/10.46471/GIGABYTE_SERIES_0002)

1  Applied Zoology and Animal Conservation Group, University of the Balearic Islands (UIB), Ctra Valldemossa km 7.5, 07122 Palma, Spain

2  Research and Innovation Centre, Fondazione Edmund Mach, Via Edmund Mach 1, 38098 San Michele all'Adige (TN), Italy

3  Kommunale Aktionsgemeinschaft zur Bekämpfung der Schnakenplage (KABS) e.V. Georg-Peter-Süß-Str. 3, 67346 Speyer, Germany

4  Austrian Agency for Health and Food Safety (AGES), Division for Public Health, Währinger Strasse 25a, 1090 Vienna, Austria

5  Laboratory of Insects & Parasites of Medical Importance, Benaki Phytopathological Institute, St. Delta 8, Kifisia 14561, Athens, Greece

6  Anticimex Spain, Carrer Jesús Serra Santamans 5 Planta 3, 08174 Sant Cugat del Vallès, Barcelona, Spain

7  Brigade Verte du Haut-Rhin, Service démoustication, 92 rue Mal. de Lattre de Tassigny, 68360 Soultz, France

8  Institute of Virology, Biomedical Research Center of Slovak Academy of Sciences, Dubravska cesta 9, 845 05 Bratislava, Slovakia

9  Agronomic and Veterinary Institute Hassan II, BP 6202, Rabat-Instituts 10100, Rabat, Morocco

10 Department of Animal Health, Veterinary Faculty, University of Extremadura, Av. de la Universidad, s/n, 10003 Cáceres, Spain

11  Dep. Public Health and Infectious Diseases, University Sapienza, Piazzale Aldo Moro 5, 00185 Roma, Italy

12  Environmental Predictions Department, Climate and Atmosphere Research Centre, The Cyprus Institute, 20 Konstantinou Kavafi Street, 2121 Nicosia, Cyprus

13  Animal Health Department, Faculty of Veterinary Medicine of Zaragoza, University of Zaragoza, C/Miguel Servet 177, 50013 Zaragoza, Spain

14  Consell Comarcal del Baix Llobregat, 08980 Sant Feliu de Llobregat, Barcelona, Spain

15  Department of Biology, Faculty of Sciences, University of Extremadura Av. de Elvas, s/n, 06006 Badajoz, Spain

16  University of Applied Sciences and Arts of Southern Switzerland, Institute of Microbiology, Vector Ecology Unit, Via Flora Ruchat-Roncati 15, 6850 Mendrisio, Switzerland

17  University of Veterinary Medicine, Institute of Parasitology, Vienna

18  NEIKER-Basque Institute for Agricultural Research and Development, Berreaga 1, 48160 Derio, Bizkaia, Spain

19  Ecodevelopment S.A., PO Box 2420, Thesi Mezaria, 57010 Filyro, Greece

20  Universidad Iberoamericana (UNIBE), Avenida Francia 129, Santo Domingo 10203, Rep. Dominicana

21  Swiss Tropical and Public Health Institute (Swiss TPH), Kreuzstrasse 2, CH-4123 Allschwil, Switzerland

22  Department of Parasitology and Parasitic Diseases, University of Agricultural Sciences and Veterinary Medicine of Cluj-Napoca, Romania

23  Centre for Monitoring of Vectors, National Reference Centre, Netherlands Food and Consumer Product Safety Authority, Geertjesweg 15, 6706 EA Wageningen, The Netherlands

24  Vectors' Control Unit, Epidemiology and Control of Infectious Diseases Department, Institute of Public Health, Rruga Aleksander Moisiu 80, Tirana, Albania

25  Faculty of Mathematics, Natural Sciences and Information Technologies, University of Primorska, Glagoljaška ulica 8, 6000 Koper, Slovenia

26  University of Novi Sad, Faculty of Agriculture, Laboratory for Medical and Veterinary Entomology, Trg Dositeja Obradovića 8, 21 000 Novi Sad, Serbia

27  Andrija Stampar Teaching Institute of Public Health, Mirogojska c. 16, 10000 Zagreb, Croatia

28  University of Pécs, Ifjúság útja 6, 7624 Pécs, Hungary

29  Bernhard Nocht Institute of Tropical Medicine, Department of Arbovirology, Hamburg, Bernhard-Nocht-Straße 74, 20359 Hamburg, Germany

30  Joint Services Health Unit, British Forces, RAF Akrotiri BFPO 57, Cyprus

31  Istituto Zooprofilattico Sperimentale delle Venezie, Viale dell'Università 10, 35020 Legnaro (Padua), Italy

32  Agencia de Salud Pública de Barcelona, Plaça Lesseps 8 entresol, 08023 Barcelona, Spain

33  Departament of Biology, Faculty of Mathematic and Natural Sciences, University of Prishtina, Str. Eqrem Qabej 9, Pristina, Republic of Kosovo

34  Cyprus University of Technology, Limassol, Archiepiskopou Kyprianou 30, Limassol 3036, Cyprus

35  National Institute of Health/ Centre for Vectors and Infectious Diseases Research, Avenida Padre Cruz, 1649-016 Lisboa, Portugal

36  Center for Rickettsiosis and Arthropod-Borne Diseases, Hospital Universitario San Pedro-CIBIR, C/Piqueras 98, 26006 Logroño, La Rioja, Spain

37  Istanbul University - Cerrahpasa, Faculty of Veterinary Medicine, Department of Parasitology, Buyukcekmece Yerleskesi, Alkent 2000 Mah, Yigitturk Cad. 5/9/1, 34500 Buyukcekmece, Istanbul, Turkey

38  University of Montenegro. Biotechnical Faculty, Mihaila Lalića 15, 81000 Podgorica, Montenegro

39  Universitat Pompeu Fabra - Mosquito Alert, C/Ramon Trias Fargas, 25-27. 08005 Barcelona, Spain

40  Friedrich-Loeffler-Institut, Suedufer 10, 17493 Greifswald Isle of Riems, Germany

41  Luxembourg National Museum of Natural History, Rue Münster 25, L-2160, Luxembourg

42  Institute for Public Health of Montenegro, bb John Jackson Street, Podgorica, Montenegro

43  Hacettepe University, Faculty of Science, Department of Biology, Ecology Section, Ankara, Turkey

44  Department of Veterinary Medicine, Faculty of Agriculture and Veterinary, University Hasan Prishtina, M546+72H, Prishtinë, Republic of Kosovo

45  PHI Center for Public Health-Skopje, blv.3rd Macedonian brigade 18, Skopje, North Macedonia

46  Centre for Ecological Research, Eötvös Lóránd Research Network, Alkotmány út 2-4, 2163 Vácrátót, Hungary

47  Institute of Zoology, Ministry of Education and Research st. Academiei 1, Chisinau MD-2028, Republic of Moldova

48  Environmental Research Group Oxford, c/o Department of Zoology, Mansfiled Road, Oxford, UK

49  Francis Schaffner Consultancy, Lörracherstrasse 50, 4125 Riehen, Switzerland

50  Agronomic and Veterinary Institute Hassan II, X4GM+H88, Rabat, Morocco

51  Institute of Public Health, Epidemiology and Control of Infectious Diseases Department, Vectors' Control Unit, Rruga Aleksander Moisiu, No. 80, Tirana, Albania

52  Universität Basel, Petersplatz 1, P.O. Box CH-4001 Basel, Switzerland

## ABSTRACT

Human and animal vector-borne diseases, particularly mosquito-borne diseases, are emerging or re-emerging worldwide. Six *Aedes* invasive mosquito (AIM) species were introduced to Europe since the 1970s: *Aedes aegypti*, *Ae. albopictus*, *Ae. japonicus*, *Ae. koreicus*, *Ae. atropalpus* and *Ae. triseriatus*. Here, we report the results of AIMSurv2020, the first pan-European surveillance effort for AIMs. Implemented by 42 volunteer teams from 24 countries. And presented in the form of a dataset named "AIMSurv Aedes Invasive Mosquito species harmonized surveillance in Europe. AIM-COST Action. Project ID: CA17108". AIMSurv2020 harmonizes field surveillance methodologies for sampling different AIMs life stages, frequency and minimum length of sampling period, and data reporting. Data include minimum requirements for sample types and recommended requirements for those teams with more resources. Data are published as a Darwin Core archive in the Global Biodiversity Information Facility- Spain, comprising a core file with 19,130 records (EventID) and an occurrences file with 19,743 records (OccurrenceID). AIM species recorded in AIMSurv2020 were *Ae. albopictus*, *Ae. japonicus* and *Ae. koreicus*, as well as native mosquito species.

**Subjects**  Ecology, Biodiversity, Taxonomy

## DATA DESCRIPTION

### Background

Vector-borne diseases (VBDs) are caused by a pathogen transmitted by vectors (often an arthropod) between hosts. Emerging or re-emerging VBDs in humans and animals are of increasing concern for public health worldwide [1], particularly mosquito-borne viral diseases such as chikungunya, dengue, West Nile fever and Zika [2, 3].

Some mosquitoes capable of transmitting pathogens are relevant invasive species at the global scale [4, 5]. They are usually introduced into new areas by global trade (for example, used tires, plants) [6–8] and have spread within Europe through human-assisted pathways favored by environmental and climate change [9–11].

In Europe, six *Aedes* invasive mosquito (AIM) species [12, 13] have been introduced since the 1970s: the yellow fever mosquito *Aedes* (*Stegomyia*) *aegypti* (Linnaeus, 1762, NCBI:txid7159); the Asian tiger mosquito, *Aedes* (*Stegomyia*) *albopictus* (Skuse, 1894, NCBI:txid7160); the Japanese bush mosquito *Aedes* (*Hulecoeteomyia*) *japonicus* (Theobald, 1901, NCBI:txid140438); the Korean bush mosquito *Aedes* (*Hulecoeteomyia*) *koreicus* (Edwards, 1917, NCBI:txid586676); the American rock pool mosquito *Aedes* (*Georgecraigius*) *atropalpus* (Coquillett, 1902, NCBI:txid28624) and the American tree-hole mosquito, *Aedes* (*Protomacleaya*) *triseriatus* (Say, 1823, NCBI:txid7162).

*Aedes aegypti* is a major vector of yellow fever, dengue and chikungunya viruses and is commonly found in tropical and subtropical areas [14]. Recent re-establishment of this species in Europe was recorded on Madeira (Portugal) [15], in parts of southern Russia, Georgia and Turkey [16–18], as well as in Fuerteventura (Canary Islands, Spain) [19], from where it was successfully eradicated in 2019. This species has also been detected at several Western European locations such as the Netherlands [20], where it has not established.

The Asian tiger mosquito originates from Southeast Asia and is currently widespread throughout large areas of Africa, Europe, Australia, the Americas, and the Middle East [14, 21]. It is one of the most invasive species in the world, according to the Invasive Species Specialist Group of the International Union for the Conservation of Nature (IUCN) [4]. In Europe, *Ae. albopictus* was first detected in Albania in 1979 [22]; nowadays this species is found in more than 27 countries. Since 2007, the Asian tiger mosquito has been linked to several outbreaks of arboviral diseases, such as dengue and chikungunya, which were introduced by travelers in different areas of Europe (for example, Italy, France, Croatia, and Spain) [5, 23]. In laboratory trials, *Ae. albopictus* is a competent vector of more than 26 arboviruses [24]. It is also a nuisance to humans, especially because of its outdoor daylight feeding behavior [25].

*Aedes japonicus* originates from eastern Asia and is established in North America, Central Europe and areas of southern Europe (such as Spain and Italy) [26–28]. It breeds in artificial containers, so its means of introduction and dispersal are like those of *Ae. albopictus* [27]. Although laboratory trials showed it to be a competent vector of West Nile virus, among others, it is not considered a major vector of VBDs in nature [29].

*Aedes koreicus* originates from Korea, Japan, and northeast China and is present in some regions of Austria, Belgium, Germany, Hungary, Italy, Slovenia and the Swiss–Italian border [30–37]. It is not considered a major vector of VBDs, although field evidence suggests it is a potential vector of Japanese encephalitis virus [38], and laboratory trials have showed low-level transmission of chikungunya and Zika viruses [39, 40]. Moreover, it is likely to be a competent vector of *Dirofilaria immitis* [41].

*Aedes atropalpus* is an invasive mosquito species originating from North America, which has been detected in European countries like Italy, France and the Netherlands without any evidence of prolonged establishment [42–45]. It is not considered to be a major vector, but in laboratory trials it is a competent vector of viruses such as West Nile [46].

Finally, *Aedes triseriatus*, which originates from North America, was detected in a single incursion in France in 2004 and was successfully eradicated [44]. La Crosse [47] and West Nile [46] viruses have been detected in field-collected mosquito adults in the USA.

## Context

Different EU initiatives exist to map the distribution of invasive mosquitoes. The European Centre for Disease Prevention and Control (ECDC) and the European Food Safety Authority (EFSA) have established VectorNet, a community network for medical entomologists and public health experts. VectorNet produces and periodically updates distribution maps of invasive mosquitoes in Europe [48]. These maps result from the analysis of literature records on the distribution of AIM species in Europe and the contribution of public, academia and research institutions that freely share their data with the VectorNet community. Another initiative is Mosquito Alert [49], a Spanish-originated citizen science campaign to monitor and map *Aedes* invasive species.



Early detection and surveillance of invasive mosquito species are challenging in terms of coordination and resources. Detection of AIM species may include different means and roles, from national surveillance programs to detect invasive species at points of entry, and to establish early warning rapid response systems to monitor AIM populations. Surveillance is commonly organized at the local and regional levels by public agencies, universities and research institutions [50], leading to different methodologies and strategies for sampling life stages of AIM (eggs, larvae/pupae and adults). Valuable guidelines for conventional surveillance have been produced by the ECDC and the World Health Organization Europe regional Office (EU-WHO) [51]. However, to date, they have never been harmonized and used simultaneously by different entomologist teams across Europe.

To increase harmonization between European entomologists, the *Aedes* Invasive Mosquito species (AIM) COST Action [52] was initiated in 2018. It had three major objectives: (i) to develop pan-European networking and collaboration in monitoring and surveillance of AIM species; (ii) to increase preparedness and capacity to fight against AIMs by triggering optimization and innovation in AIM control strategies; and (iii) to disseminate, customize and communicate the AIM-COST Action outcomes.

The AIM-COST Action aims to promote data sharing and harmonization. A particularly important objective is to ensure that vector sampling is consistent and compatible throughout Europe to enable an accurate continental picture of vector distributions. For this, AIM-COST organized a training course in Cyprus in January 2020 on harmonizing AIM surveillance across Europe. As a result of this course, trainers and trainees developed a protocol for surveying AIM species that can be applied across Europe. Forty-two teams from 24 countries (23 from Europe and one from North Africa) agreed to participating in the first ever pan-European surveillance of AIMs using a harmonized protocol [53]. The AIMSurv protocol was first implemented in 2020, then extended to 2021 and 2022. The main aim was to provide longitudinal data enabling comparison of seasonality and abundance across Europe and, in a subsequent phase, to compare field data with reports obtained by citizen science (for example, the Mosquito Alert App [54], the data of which has also been published in GBIF [55]). Accordingly, both the presence and absence results of AIMs species were considered equally important to improve the information at the continental level.

## METHODS

The sampling protocol for pan-European surveillance of AIM species (AIMSurv) harmonized the sampling methods, frequency, minimum length of the sampling period and the form of reporting. A minimum requirements protocol (MRP) was established for different samples (for example, eggs in ovitraps), number of sampled sites, number of traps and frequency of collecting samples. For teams with more resources, a recommended protocol (RP) was also established to either increase the number of samples and/or to sample life stages other than eggs, such as adults.

The use of a common platform for data collection was also suggested: the VECMAP® App system [56] (Avia-GIS, Zoersel, Belgium), which was made freely available by Avia-GIS to all participants during AIMSurv activities.

### Minimum requirements protocol

For the MRP, all teams performed the survey in three sampling sites separated by 10 km or more. Five oviposition traps (ovitraps) per site were placed and separated by 15–100 m.

The type of ovitrap was selected by each team according to their availability in the region, but usually consisted of 250 to 1000-ml capacity black containers filled with tap water. One scratched wooden tongue depressor (1.7 × 15 cm) per ovitrap was used as a substrate for oviposition. Some teams used similar sized pieces of Masonite board (when part of a pre-existing surveillance network was in place).

The selected sampling sites shared a similar ecology, when possible, in urban and/or suburban areas (e.g., a garden of single-family houses in residential urban/suburban areas, public parks near residential areas, recreational areas). The frequency of sample collection was biweekly over a minimum of 3 months, which included the population peak of the targeted AIM species (e.g., in Spain: from September to November for *Ae. albopictus*).

The following parameters were recorded: latitude and longitude of the position of each trap; the name of municipality/county/district (according to the country) and locality; start and end date of each trapping event (e.g., a period of 14 days for ovitraps); land use categories (urban, suburban and others); count of each life stage collected (egg and adult), including absences (0 values).

## Recommended protocol

The more ambitious RP sampling included additional sampling sites sampled by five ovitraps per site, weekly sampling frequency and sampling length during the whole seasonality of the AIM species including start, peak and end of the mosquito season (e.g., May to November in Central Europe for *Ae. albopictus*). In addition, sampling adults using one BG-Sentinel™ (Biogents, Germany) trap baited with BG-Lure™ (Biogents, Germany) and/or $CO_2$ per site under a sampling frequency of one trap/night per week was also included. The use of VECMAP® (AVIA-GIS, Belgium) to report the data was also suggested in the RP. Parameters to record were the same as for the MRP, plus the daily or weekly record of meteorological parameters (maximum, minimum, average temperature) per site, collected using data loggers or local weather stations (data not included in the current dataset).

The trap status per trapping event was recorded as follows: 'Valid' when the trap (either oviposition or BG-Sentinel) was fully functional during the sampling event; 'Trap altered' when oviposition trap was found dry or turned over, or objects or animals, such as snails and lizards, were found inside, but the sample could still be collected. 'Trap altered' also referred to BG-Sentinel traps when they were found unplugged or with the battery switched off, or if the funnel was blocked, but the sample could still be collected.

To process samples, eggs of AIM species collected were counted. When needed, for every location a subsample (two out of five ovitrap substrates per locality) of eggs was reared to confirm the species by larva/adult morphology, particularly in areas where several AIM species are present (i.e., *Ae. albopictus* and *Ae. japonicus*). Alternatively, when possible and depending on the team's resources, species were identified using: matrix-assisted laser desorption ionisation–time of flight mass spectrometry (MALDI-TOF MS) or molecular methods (e.g., DNA sequencing).

Adults of AIM species collected in BG-Sentinel™ traps were identified by morphology, and sexed and counted. Suggested identification keys were ECDC (2012) [51] and MosKeyTool V2.1 [57]. Samples of adults were preserved in 96% ethanol and/or cold preserved at −20/−80 °C to confirm identification if needed (e.g., via molecular tools).

## DATA VALIDATION AND QUALITY CONTROL

All participants in AIMSurv reported data using a harmonized template. All data reported has been curated and the terminology has been homogenized. Data has been validated using the validator available at the Global Biodiversity Information Facility (GBIF) [58].

## REUSE POTENTIAL

Records presented here represent the first pan-European data on field surveillance of AIM species conducted with harmoniously methodologies and time scales across 24 countries. The records allow the accurate comparison of AIM surveillance, abundance, and seasonality between countries and/or regions. Data can also be compared with other sampling strategies of AIM species, such as citizen science.

## DATA AVAILABILITY

The data supporting this article are published through the Universitat de les Illes Balears IPT and are available under a CC0 waiver from GBIF [58]. We kindly ask users to give appropriate credit and attribution if you use this data.

## EDITOR'S NOTE

This paper is part of a series of Data Release articles working with GBIF and supported by the Special Programme for Research and Training in Tropical Diseases (TDR), hosted at the World Health Organization [59].

## DECLARATIONS
## LIST OF ABBREVIATIONS

AIM: Aedes invasive mosquito; COST: European Cooperation in Science and Technology; ECDC: European Centre for Disease Control; EFSA: European Food Safety Authority; EU: European Union; EU-WHO: World Health Organization Europe regional office; IUCN: International Union for the Conservation of Nature; MALDI-TOF MS: matrix-assisted laser desorption ionization–time of flight mass spectrometry; MRP: minimum requirements protocol; RP: recommended protocol; VBD: vector-borne disease

## ETHICAL APPROVAL

Not applicable.

## CONSENT FOR PUBLICATION

Not applicable.

## COMPETING INTERESTS

The authors declare that they have no competing interests.

## FUNDING

This study was funded by the Autonomous Province of Trento (Italy) under the project 'Coordinated surveillance actions on invasive alien species and emerging vector borne diseases'; the City Health Office of the City of Zagreb, within the 'Program for monitoring invasive mosquito species in the area of the City of Zagreb in 2020'; the Consejería de Economía e Infraestructura of the Junta de Extremadura and the European Regional Development Fund, a Way to Make Europe, through the research project IB16135; Dirección



de Salud Pública (Gobierno Vasco), Project EU-LIFE 18 IPC/ES/000001 (Urban Klima 2050) y Programa Estatal de Vigilancia de mosquitos en puertos y Aeropuertos, del Ministerio de Sanidad (Gobierno de España); EMME-CARE project, which has been funded from the European Union's Horizon 2020 Research and Innovation Programme (grant agreement ID 856612); Institute of Zoology under the project EVOLANTER (project no. 20.80009.7007.02). RL is funded by the Federal Ministry of Education and Research of Germany (BMBF) under the project NEED (grant no. 01Kl2022); LIFE CONOPS project (LIFE12 ENV/GR/000466), funded by the European Commission in the framework of the program LIFE + Environment Policy and Governance; Municipalities of Slovenia: City Municipality of Nova Gorica, City Municipality of Koper, Municipality of Izola, Municipality of Piran and Municipality of Ankaran; National Research, Development and Innovation Office (NKFIH grant numbers KH-130379, PD-135143, FK-138563 and K-135841). The research activity of KK was supported by the Janos Bolyai Research Scholarship of the Hungarian Academy of Sciences and by the ÚNKP-20-5-PTE-597 New National Excellence Program of the Ministry for Innovation and Technology; Portuguese National Program for Vector Surveillance (REVIVE) and we are particularly grateful to the regional workgroup of Algarve for the monitoring activities; PR (19_ECO_0070) project 'Aves y Enfermedades Infecciosas Emergentes: Impacto de las Especies Exóticas y Migratorias en la transmisión de Malaria aviar y el virus del Nilo Occidental – AvEIEs' from 'Ayudas Fundación BBVA a Equipos de Investigación Científica 2019'; project grant number 57 PCCDI/2018, grant agency 'The Executive Unit for Financing Higher Education, Research, Development and Innovation' (UEFISCDI) Romania, 'Collegium Talentum Programme' of Hungary, Eötvös Loránd University's 'Homeland higher education study grant'; Slovak Research Agency VEGA nr. 2/0140/21; Vector Control Needs Assessment in Cyprus, contracted by the World Health Organization (reference 2020/1040069-0); Veneto and Friuli Venezia Giulia Regions (Regional Prevention Plans 'Entomological Surveillance of vector-borne diseases' in the Veneto and Friuli Venezia Giulia Regions); the Institute of Public Health, Albania under the program of mosquitoes control in urban and coastal areas.

## AUTHORS' CONTRIBUTIONS

MAM and AdT conceived this work; MAM, AM, WW, DP and FS designed this work; CB, DA, XA, KB-L, GB, MB, PB, KB, MB, DB-B, VČ, BC, MC, SD, RE, OF-F, MF, EF, EMF-C, HPF, ALG-P, PG, SG, FG, MAG, MG, RG-L, CH, AI-J, VJ, PK, KK, MK, AK, KK, JL, RL, SM, GM, AFM, AM, AM, TM, FM, SM, NM-B, PM, GN, HCO, JAO, KO, IP, JRBP, SP, CR, CR, ER, IR-A, IS-C, NS, KS, KS, CS, MS, NiS, ZS, TS, JŠ, ST, AV, MIV, EV, AM, DP, FS collected the samples and reported results; MAM wrote the original draft and coordinated AIMSurv data compilation and curation; AdT coordinated AIM-COST Action. All authors read, revised, and approved the final manuscript.

## ACKNOWLEDGEMENTS

We acknowledge the support provided by Katia Cezón (GBIF-Spain) in adapting our dataset to the GBIF standards. The work was done within the framework of AIM-COST Action (CA17108).

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

## DETAILS OF COLLABORATIVE AUTHORS

### • List of authors in Consortium AIM-COST/AIM-Surv

Carina Suchentrunk,[31] Thomas Zechmeister,[31] Elfriede Gruber,[37] Gerd Orehounig,[37] Grete Altgayer,[37] Franz Lex,[34] Inge Lebl,[38] David Zezula,[41] Jana S. Petermann,[41] Florian Oberleitner,[43] Carina Zittra,[24] Thomas Brenner,[35] Klaus Zimmermann,[39] Lisa Klocker,[39] Barbara Eigner,[1] Licha Wortha,[32] Stephanie Pree,[32] Stefanie Jäger,[40] Thorsten Schwerte,[40] Christian Wieser,[33] Helge Heimburg,[33] Johana Gunczy,[36] Wolfgang Paill,[36] Hans Jerrentrup,[42] S. Pree,[1] E. Daroglou,[42] B. Eigner,[32] B. Shahi-Barogh,[32] L.N. Wortha,[1] Marek Svitok,[2] Ivana Svitková,[2] Jozef Oboňa,[2] Eva Barbušinová,[2] Martina Micocci,[3] Marta Albani,[3] Paola Serini,[3] P. Cobre,[3] Moisès Canals,[4] Roser Bellés,[4] Kamil Erguler,[5] Marco Neira,[5] Nikolaos Kelemenis,[6] Giorgios Vlachos,[6] Antonis Karagiannis,[6] Jesús F. Barandika,[7] Aitor Cevidanes,[7] Patricia Vázquez,[7] Arjan Stroo,[8] Zala Horvat,[9] Maša Stranj,[9] A. Ignjatović-Ćupina,[10] D. Dondur,[10] S. Bogdanović,[10] V. Srdić,[10] Z. Francuski,[10] A. Žunić,[10] Marcela Curman Posavec,[11] Danijel Poje,[11] Tomislav Pismarovic,[11] G. Markó,[12] Enrico Inama,[13] Mattia Manica,[13] Annapaola Rizzoli,[13] K. Athanasiou,[14] A. Muja,[15] H. Qollaku,[15] Fátima Amaro,[16] Nélia Guerreiro,[16] B. Alten,[28] F. Gunay,[28] O.Y. Eryigit,[29,30] B. Yildirim,[29] S.O. Yilmaz,[29] S. Pehlivan,[17] U. Neumann,[18] O. Tauchmann,[18] A. Vasic,[18] Galina Busmachiu,[19] U. Lange,[25] J. Schmidt-Chanasit,[25] I Angelidou,[20] C. Panayiotou,[20] I. Konstantinou,[20] Gj. Sino,[21] Haki Mema,[26] Altin Veliko,[27] Dimitra Kollia,[22] Fotis Mourafetis,[22] Vasileios Karras,[22] Marina Bisia,[22] Christelle Bender[23]

[1] Austrian Agency for Health and Food Safety (AGES), Division for Public Health, Währinger Strasse 25a, 1090 Vienna, Austria

[2] Institute of Virology, Biomedical Research Center of Slovak Academy of Sciences, Dubravska cesta 9, 845 05 Bratislava, Slovakia

[3] Dep. Public Health and Infectious diseases, University Sapienza, Piazzale Aldo Moro 5, 00185 Roma, Italy

[4] Consell Comarcal del Baix Llobregat, 08980 Sant Feliu de Llobregat, Barcelona, Spain

[5] Environmental Predictions Department, Climate and Atmosphere Research Centre, The Cyprus Institute, 20 Konstantinou Kavafi Street, 2121, Nicosia, Cyprus

[6] Ecodevelopment S.A., PO Box 2420, Thesi Mezaria, 57010 Filyro, Greece

[7] NEIKER-Basque Institute for Agricultural Research and Development, Berreaga 1, 48160 Derio, Bizkaia, Spain

[8] Centre for Monitoring of Vectors, National Reference Centre, Netherlands Food and Consumer Product Safety Authority, Geertjesweg 15, 6706 EA Wageningen, The Netherlands

[9] Faculty of Mathematics, Natural Sciences and Information Technologies, University of Primorska, Glagoljaška ulica 8, 6000 Koper, Slovenia

[10] University of Novi Sad, Faculty of Agriculture, Laboratory for Medical and Veterinary Entomology, Trg Dositeja Obradovića 8, 21 000 Novi Sad, Serbia

[11] Andrija Stampar Teaching Institute of Public Health, Mirogojska c. 16, 10000 Zagreb, Croatia

[12] Szent István University, Budapest, Hungary

[13] Research and Innovation Centre, Fondazione Edmund Mach, Via Edmund Mach 1, 38098 San Michele all'Adige (TN), Italy

[14] Joint Services Health Unit, British Forces, RAF Akrotiri BFPO 57, Cyprus

[15] Departament of Biology, Faculty of Mathematic and Natural Sciences, University of Prishtina, Str. Eqrem Qabej 9, Pristina, Republic of Kosovo

[16] National Institute of Health/ Centre for Vectors and Infectious Diseases Research, Avenida Padre Cruz, 1649-016 Lisboa, Portugal

[17] Istanbul University - Cerrahpasa, Faculty of Veterinary Medicine, Department of Parasitology, Buyukcekmece Yerleskesi, Alkent 2000 Mah., Yigitturk Cad. 5/9/1, 34500 Buyukcekmece, Istanbul, Turkey

[18] Friedrich-Loeffler-Institut, Suedufer 10, 17493 Greifswald Isle of Riems, Germany

[19] Institute of Zoology, Ministry of Education and Research st. Academiei 1, Chisinau MD-2028, Republic of Moldova

[20] Cyprus University of Technology, Limassol, Archiepiskopou Kyprianou 30, Limassol 3036, Cyprus

[21] Vectors' Control Unit, Epidemiology and Control of Infectious Diseases Department, Institute of Public Health, Rruga Aleksander Moisiu 80, Tirana, Albania

[22] Laboratory of Insects & Parasites of Medical Importance, Benaki Phytopathological Institute, 8 Stefanou Delta str, Kifisia 14561, Athens, Greece

[23] Syndicat de lutte contre les moustiques du Bas-Rhin, 19-21 rue de la premire arme, 67630 Lauterbourg, France

[24] University of Vienna, Dept. of Functional & Evolutionary Ecology, Djerassiplatz 1, 1030 Vienna, Austria

[25] Bernhard Nocht Institute of Tropical Medicine, Department of Arbovirology, Hamburg, Bernhard-Nocht-Straße 74, 20359 Hamburg, Germany

[26] Local Health Care Unit Fier, Albania

[27] Local Health Care Unit Vlore, Albania

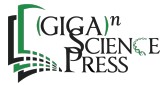

[28] Hacettepe University, Faculty of Science, Department of Biology, Ecology Section, Ankara, Turkey

[29] Istanbul Metropolitan Municipality, Head of Health Department, Istanbul, Turkey

[30] Istanbul Istun Health and Technology University, School of Health, Istanbul, Turkey

[31] Biological Station Lake Neusiedl, Illmitz, Austria

[32] Institute of Parasitology, University of Veterinary Medicine, Vienna, Austria

[33] Landesmuseum Kärnten, Klagenfurt, Austria

[34] Citizen Scientist, Neuhaus am Klausenbach, Austria

[35] GEBL – Gelsenbekaempfung Leithaauen, Mannersdorf, Austria

[36] Universalmuseum Joanneum, Graz, Austria

[37] Citizen Scientist, Althofen, Austria

[38] Citizen Scientist, Vienna, Austria

[39] inatura GmbH, Dornbirn, Austria

[40] Institute of Zoology, University of Innsbruck, Austria

[41] Department of Biosciences, University of Salzburg, Austria

[42] Verein Biologische Gelsenregulierung in den March-Thaya-Auen, Hohenau, Austria

[43] Department of Ecology, University of Innsbruck, Austria

