## [Reviewer Report]

Upload additional filesDRR-202202-03/form/DRR-202202-03-Data-Review.pdfReviewer name and names of any other individual's who aided in reviewer Christopher HunterDo you understand and agree to our policy of having open and named reviews, and having your review included with the published papers. (If no, please inform the editor that you cannot review this manuscript.)YesIs the language of sufficient quality?YesPlease add additional comments on language quality to clarify if needed
Are all data available and do they match the descriptions in the paper? YesAdditional CommentsAre the data and metadata consistent with relevant minimum information or reporting standards? See GigaDB checklists for examples <a href="http://gigadb.org/site/guide" target="_blank">http://gigadb.org/site/guide</a>YesAdditional CommentsNB- the GBIF dataset is currently listed as CC-BY, this needs to be changed to CC-0Is the data acquisition clear, complete and methodologically sound?YesAdditional CommentsIs there sufficient detail in the methods and data-processing steps to allow reproduction?YesAdditional CommentsIs there sufficient data validation and statistical analyses of data quality? YesAdditional CommentsIs the validation suitable for this type of data?YesAdditional CommentsIs there sufficient information for others to reuse this dataset or integrate it with other data?YesAdditional CommentsAny Additional Overall Comments to the AuthorThe datanote described the coalition of multiple european partners coming together to survey and monitor the distribution of mosquitoes across europe. It is well written and describes the context and collection methods very well.
In general, the GBIF dataset accompanying this manuscript is well annotated and very close to complete for the time period covered.
The manuscript alludes to additional data being collected for at least part of the study “The more ambitious RP sampling included…the daily or weekly record of meteorological parameters (maximum, minimum, average temperature) per site, collected using data loggers or local weather stations.” Those data are not available in the GBIF dataset. Are they available elsewhere? If so, can a PID be provided for access to those data? If they are not currently hosted publicly anywhere, then a GigaDB dataset can be created to host them.

Major comments :
1 - The GBIF dataset needs to be made CC-0
2 - provide access to the additional data collected for the RP sampling

Minor comments:
1 - if the missing GPS coordinates are available they should be added to the GBIF dataset.
2 - consider adding more than 1 contributor to the GBIF dataset
See data review document attached additional information.
RecommendationMinor Revision

---

## [Reviewer Report]

Reviewer name and names of any other individual's who aided in reviewer Frank W. AvilaDo you understand and agree to our policy of having open and named reviews, and having your review included with the published papers. (If no, please inform the editor that you cannot review this manuscript.)YesIs the language of sufficient quality?YesPlease add additional comments on language quality to clarify if needed
Are all data available and do they match the descriptions in the paper? YesAdditional CommentsAre the data and metadata consistent with relevant minimum information or reporting standards? See GigaDB checklists for examples <a href="http://gigadb.org/site/guide" target="_blank">http://gigadb.org/site/guide</a>YesAdditional CommentsIs the data acquisition clear, complete and methodologically sound?YesAdditional CommentsThe data acquisition methods are very thorough and well explained. Is there sufficient detail in the methods and data-processing steps to allow reproduction?YesAdditional CommentsIs there sufficient data validation and statistical analyses of data quality? Not my area of expertiseAdditional CommentsIs the validation suitable for this type of data?YesAdditional CommentsIs there sufficient information for others to reuse this dataset or integrate it with other data?YesAdditional CommentsAny Additional Overall Comments to the AuthorRecommendationAccept

---

## [Reviewer Report]

Upload additional filesDRR-202202-03/form/GBIF_draft_22_03_Reviewed (1).docxReviewer name and names of any other individual's who aided in reviewer Andrés Carrazco-MontalvoDo you understand and agree to our policy of having open and named reviews, and having your review included with the published papers. (If no, please inform the editor that you cannot review this manuscript.)YesIs the language of sufficient quality?YesPlease add additional comments on language quality to clarify if needed
Are all data available and do they match the descriptions in the paper? YesAdditional CommentsAre the data and metadata consistent with relevant minimum information or reporting standards? See GigaDB checklists for examples <a href="http://gigadb.org/site/guide" target="_blank">http://gigadb.org/site/guide</a>YesAdditional CommentsIs the data acquisition clear, complete and methodologically sound?NoAdditional CommentsIs there sufficient detail in the methods and data-processing steps to allow reproduction?YesAdditional CommentsIs there sufficient data validation and statistical analyses of data quality? YesAdditional CommentsIs the validation suitable for this type of data?YesAdditional CommentsIs there sufficient information for others to reuse this dataset or integrate it with other data?NoAdditional CommentsAny Additional Overall Comments to the AuthorRecommendationMinor Revision